# Quantum entanglement for attention models

## Abstract

Attention mechanisms in deep learning establish relationships between different positions within a sequence, enabling models like Transformers to generate effective outputs by focusing on relevant input segments and their relations. The performance of Transformers is highly dependent on the chosen attention mechanism, with various approaches balancing trade-offs between computational cost, memory efficiency, and generalization ability based on the task. Quantum machine learning models possess the potential to outperform their classical counterparts in specialized settings. This makes exploring the benefits of quantum resources within classical machine learning models a promising research direction. The role of entanglement in quantum machine learning, whether in fully quantum or as subroutines in classical-quantum hybrid models, remains poorly understood. In this work, we investigate the hypothesis of whether entanglement can be used to model nuanced correlations in classical data, analogous to its role in many-body systems. We further test whether quantum entanglement can be used as a resource to improve the performance of the attention layer in Transformers. We introduce an entanglement entropy-based attention layer within a classical Transformer architecture and numerically evaluate it across various datasets. Our experiments on standard classification tasks in both vision and NLP domains reveal that the entanglement-based attention layer outperforms existing quantum attention frameworks and the widely used quantum kernel attention models, particularly in the presence of noise. Our work contributes toward exploring the power of quantum resources as a subroutine in the classical-quantum hybrid setting to further enhance classical models.

## 1 Introduction

Machine learning has significantly affected numerous domains by enabling systems to learn complex patterns from large datasets. This progress is largely due to deep neural networks (DNNs), architectures inspired by the human brain comprising interconnected layers of artificial neurons. Within this paradigm, convolutional neural networks (CNNs) have excelled at processing grid-like data, such as images and time series, due to their inherent capability to capture local patterns and hierarchical features. However, CNNs face limitations when addressing sequential data tasks requiring modeling long-range dependencies and broader contextual relationships.

To overcome these limitations, Transformers (Vaswani et al., 2017) were developed, introducing an attention mechanism to selectively focus on relevant input segments. Initially proposed for natural language processing (NLP) tasks, Transformers have subsequently been adapted to domains such as computer vision (Dosovitskiy et al., 2020; Carion et al., 2020), audio processing (Dong et al., 2018), and numerous other domains. The core strength of Transformers lies in their attention mechanism, which quantifies the importance of inputs through attention weights, capturing intricate relationships within sequential data effectively.

In a seemingly unrelated world, physicists employ quantum mechanical wave functions to model complex interactions within systems of particles to describe the system accurately. Such quantum systems inherently exhibit entanglement, a quantum-specific correlation between particles, which classical models cannot adequately represent. Quantum entanglement, quantified using measures such as entanglement entropy, captures deep correlations within quantum systems, making it an indispensable tool in quantum mechanics.

Intriguing parallels between quantum mechanical systems and deep neural networks have been explored by (Levine et al., 2017; 2019), who demonstrated structural equivalence between them through representations of the tensor network (TN). Specifically, they observed that the expressive power of neural networks and quantum wave functions share commonalities in their ability to model intricate correlations within high-dimensional inputs. These insights suggest that quantum-inspired measures, such as entanglement entropy, might enhance the modeling capabilities of classical neural networks by introducing deeper correlation metrics.

Inspired by these parallels, we hypothesize that integrating quantum entanglement measures as attention mechanisms into classical models could better capture nuanced correlations within sequential data. Previous studies, such as Attention-based Quantum Tomography (AQT) (Cha et al., 2021) and Quantum-aware Transformers (QAT) (Ma et al., 2023), successfully leveraged attention mechanisms to reconstruct quantum states, further motivating our exploration of the reverse direction: incorporating quantum-inspired measures of correlation–entanglement–as attention mechanism of Transformer model for classical data modeling tasks.

Despite theoretical advantages suggested by quantum machine learning (QML) studies (Servedio & Gortler, 2001; Liu et al., 2021; Gyurik & Dunjko, 2023; Jäger & Krems, 2023; Molteni et al., 2024), practical implementations have shown mixed results, often failing to surpass classical baselines (Bowles et al., 2024). Thus, a gap remains concerning the utility and added value of quantum and quantum-inspired techniques in real-world tasks.

To address this gap, we propose a novel Transformer encoder model that replaces the traditional dot-product attention with attention coefficients derived from quantum entanglement entropy computed via parameterized quantum circuits (PQC). Our methodology consists of three primary steps:

1. **Quantum Embedding:** Classical *query* and *key* vectors of Transformers are encoded into quantum states using a Quantum Feature Map (QFM).

2. **Entanglement of Quantum States:** Applying a Parameterized Quantum Circuit (PQC) to entangle these quantum states.

3. **Measurement of Entanglement Entropy:** Computing entanglement entropy between *query* and *key* quantum states to estimate attention coefficients.

Our experiments demonstrate that the proposed entanglement-based attention mechanism performs comparably to classical scaled-dot product attention (Vaswani et al., 2017) on most datasets and outperforms quantum-inspired models based on the swap test, particularly under noisy conditions. Our results also suggest that the entanglement-based attention mechanism performs better in the data-limited regime compared to classical scaled-dot product attention, a feature highly desirable in a wide range of real-world applications where available data is scarce.

In summary, our work contributes a novel integration of quantum-inspired methodologies into classical Transformer models, providing new insights into modeling complex correlations within sequential data. We provide a detailed description of the methodology, experiments, and results obtained in the subsequent sections.

## 2 Related work

El Amine Cherrat et al. (2022) introduced a Quantum Vision Transformer for classification tasks on MNIST datasets. Their approach efficiently handled quantum matrix multiplications but did not demonstrate significant advantages or superior performance compared to their classical counterparts.

Khatri et al. (2024) recently proposed Quixer, a Transformer model leveraging the Linear Combination of Unitaries (Childs & Wiebe, 2012) and Quantum Singular Value Transform (Gilyén et al., 2019) for attention computations. Tested on the Penn Treebank dataset, Quixer showed competitive performance against classical baselines. Similarly, the SASQuaTCh architecture (Evans et al., 2024) implements self-

attention through the Quantum Fourier Transform in a purely quantum setting, although the absence of comparative analysis limits understanding of its practical advantages.

The Quantum Self-Attention Neural Network (QSANN) by Li et al. (2022) employs a Gaussian-projected quantum self-attention mechanism and demonstrated superior performance in text classification tasks compared to existing Quantum Natural Language Processing (QNLP) models (Lorenz et al., 2023). In this work, we include QSANN as a baseline for comparison.

Zhao et al. (2023) introduced the Quantum Kernel Self-Attention Network (QKSAN) using the Deferred Measurement Principle (DMP) and conditional measurement techniques. Their experiments, conducted on binary classification tasks for subsets of MNIST and Fashion MNIST datasets, indicated limited success, achieving around 10% accuracy for multi-class scenarios. In contrast, our work evaluates entanglement-based attention on multi-class datasets, including complete MNIST, FMNIST, and MNIST-1D, aiming to provide a more comprehensive evaluation.

Several quantum Transformer models proposed in recent literature are predominantly theoretical with limited empirical analyses. For example, the Grover-inspired Quantum Hard Attention Network (GQHAN) (Zhao et al., 2024) and Quantum Algorithm for Attention Computation (Gao et al., 2023) integrate Grover's algorithm into attention mechanisms, but lack rigorous practical evaluations. Similarly, Liao & Ferrie (2024) and Guo et al. (2024) designed quantum circuits implementing adapted Transformer components and generative pretraining, yet practical comparisons with classical counterparts are limited.

Shi et al. (2023) proposed quantum dot-product computation methods mapping query and key vectors to quantum states, evaluating on the MC and RP datasets. Shi et al. (2022) developed the Quantum Self-Attention Network (QSAN), employing Quantum Logic Similarity (QLS) and Quantum Bit Self-Attention Score Matrices (QBSASM), with evaluations limited to binary classification on simplified MNIST tasks. Di Sipio et al. (2022) explored replacing linear layers to generate query, key, and value vectors with Parameterized Quantum Circuits (PQCs), yet they provided no empirical validation.

In contrast to previous studies, we introduce a novel attention mechanism based on quantum entanglement measures. Specifically, our approach integrates the entanglement entropy computed via PQCs directly into the attention mechanism. To our knowledge, this study is the first to leverage entanglement entropy within classical machine learning, demonstrating its potential to enhance correlation modeling beyond conventional quantum-inspired and classical techniques.

## 3 Attention mechanism in transformers

Transformers typically employ an encoder-decoder structure consisting of stacked attention and fully connected layers, combined with layer normalization and residual connections (Vaswani et al., 2017). The attention layer specifically enables the model to relate different parts of an input sequence, effectively computing representations by assigning varying importance to individual segments. A single-head self-attention mechanism can be described as follows. Given an input matrix $Z \in \mathbb{R}^{N \times d}$, representing $N$ tokens each of dimensionality $d$, we first generate query, key, and value vectors using learnable parameters:

$$Q = W_q Z^\top, K = W_k Z^\top, V = W_v Z^\top \in \mathbb{R}^{d \times N}, \tag{1}$$

$$A = QK^\top \in \mathbb{R}^{N \times N}, \tag{2}$$

$$\text{Attention}(Z) = \text{softmax}(A/\sqrt{d_h})V^\top \in \mathbb{R}^{N \times d}, \tag{3}$$

where $W_q, W_k$, and $W_v \in \mathbb{R}^{d \times d}$ are the learnable query, key, and value weight matrices, respectively. Note that we do not apply output projection $W_o$ as we only consider single attention head. Here, the attention coefficient matrix $A$ represents pairwise dot-product similarities between query and key vectors. This dot product similarity metric quantifies relationships between input tokens. In this work, we aim to replace the classical dot product computation with a quantum-inspired metric based on entanglement entropy.

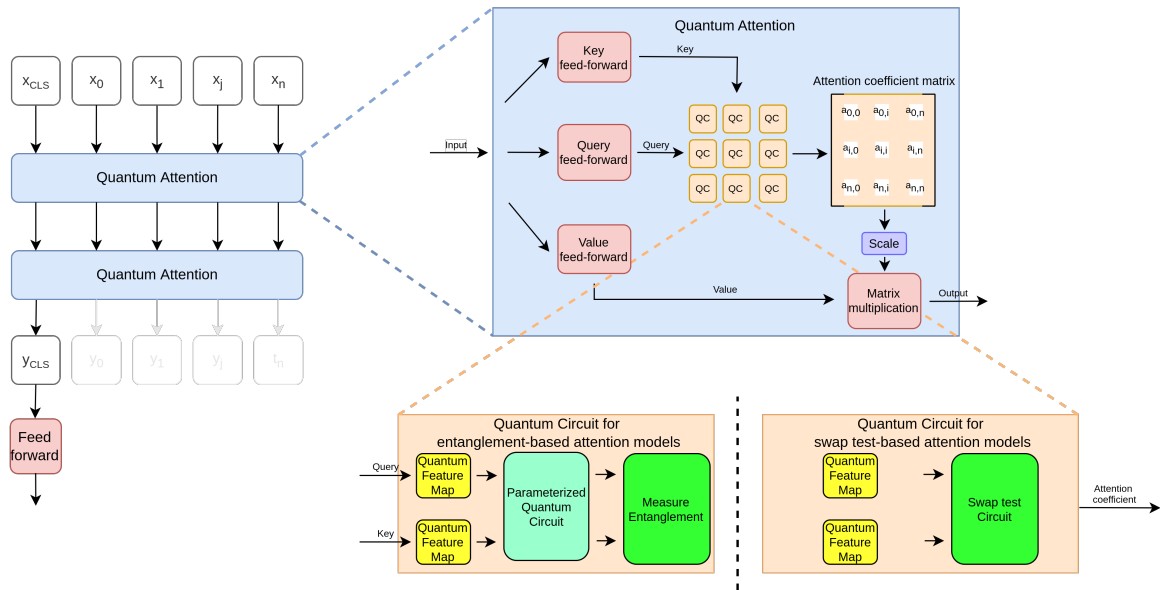

Figure 1: **Schematic of classical-quantum network architecture proposed in this work.** It is based on the Transformer encoder architecture, consisting of two sequential attention layers and a feed-forward layer. The input is embedded with a class (CLS) token denoted as $x_{\mathrm{CLS}}$, which is used to classify each sample. After the second attention layer, all tokens except the CLS token are discarded, and only the CLS token ($y_{\mathrm{CLS}}$) is passed to the feed-forward layer. For classical attention (used as a baseline), the dot product between the query and key vector serves as the attention coefficient. For entanglement-based attention, the query and key vector are encoded as a quantum state using a Quantum Feature Map (QFM) (see Section 4.1) and then entangled using a Parameterized Quantum Circuit (PQC) (see Section 4.2). The QFM employs `RX`, `RY`, `RZ` and `RZZ` gates, while the PQC utilizes `controlled-RX` gates. The entanglement entropy between the query and key states is subsequently used as the attention coefficient (see Section 4.3). In contrast, for swap test-based attention (see Section 5), the query and key vectors are encoded as quantum states using a Quantum Feature Map (QFM). The quantum state similarity is then computed via a swap test circuit, with the resulting similarity score serving as the attention coefficient.

## 4 Entanglement-based attention

We propose an entanglement-based attention mechanism, aiming to leverage quantum entanglement to capture complex relationships within datasets, analogous to its role in modeling quantum particle interactions. Our approach integrates quantum entanglement directly into the attention mechanism of a Transformer model. Similarly to classical attention, we initially compute key, query, and value vectors using feedforward layers. The subsequent quantum integration proceeds as follows:

### 4.1 Quantum embedding

Quantum computers inherently encode data into quantum states within a complex, high-dimensional Hilbert space. To take advantage of this representation, Quantum Feature Maps (QFMs) encode classical data into quantum states. QFMs associate classical feature values with parameters that control quantum gates, preparing the corresponding quantum states. We investigate three distinct quantum embedding techniques in our experiments:

1. **Super-dense angle encoding:** Each qubit encodes four classical features using a gate sequence of `RX-RY-RX-RY`, requiring one qubit per four classical features.

2. **Dense angle encoding:** Each qubit encodes two classical features using a gate sequence of `RX-RY`, effectively requiring one qubit per two classical features.

3. **Instantaneous quantum polynomial-time (IQP) encoding:** Introduced by Havlicek et al. (2018), it uses diagonal gates (`RZ`, `RZZ`) following Hadamard transformations, with one qubit per classical feature. IQP encoding provides efficient data representation and potential quantum speedups.

The circuit schematics of the techniques discussed above are depicted in the Appendix A.

## 4.2 Entangling quantum states

After embedding query and key vectors as quantum states, we apply a Parameterized Quantum Circuit (PQC) to entangle these states. PQCs have tunable parameters optimized for specific tasks, and their ability to generate entanglement is crucial for capturing correlations. The ability of a parameterized quantum circuit (PQC) to produce entanglement, often quantified using the Meyer–Wallach entanglement metric (Meyer & Wallach, 2002), is referred to as its entangling capability. This property has been widely examined across different PQC architectures (Sim et al., 2019; Hubregtsen et al., 2021). We use `controlled-RX` gates exclusively in our PQC to entangle query-key state pairs, emphasizing entanglement generation. Single-qubit gates are excluded, as they do not contribute to entanglement. Inspired by data reuploading (Pérez-Salinas et al., 2020), we iteratively apply QFM and PQC multiple times, enhancing the circuit expressivity to capture higher-order correlations.

## 4.3 Entanglement measure as attention coefficients

Entanglement generated by the PQC is quantified using a measure of entanglement (ME) to compute the attention coefficient matrix. Specifically, for each query-key pair:

$$A = \text{ME}\left(U_{\text{PQC}}(|\phi\rangle_{\text{query}} \otimes |\phi\rangle_{\text{key}})\right), \tag{4}$$

where $U_{\text{PQC}}$ denotes the unitary operation applied by the PQC. The matrix $A$ represents entanglement between each query-key pair, serving as quantum-inspired attention coefficients. Two entanglement measures are considered in this work:

1. **Von Neumann entanglement entropy:** The Von Neumann entanglement entropy for the subsystem *query* is defined as: $S_{\text{query}} = -\text{Tr}[\rho_{\text{query}} \log(\rho_{\text{query}})]$ where $\rho_{\text{query}} = \text{Tr}_{\text{key}}(|\Psi\rangle\langle\Psi|)$ is the reduced density matrix obtained by tracing out the *key* subsystem from the overall state $|\Psi\rangle = U_{\text{PQC}}(|\phi\rangle_{\text{query}} \otimes |\phi\rangle_{\text{key}})$. Calculating the Von Neumann entropy using classical shadows (Huang & Kueng, 2019; Vermersch et al., 2024) requires a number of measurements that scales quadratically with the desired precision, but notably, this measurement count is independent of system size.

2. **Rényi's second-order entanglement entropy:** The Rényi second-order entanglement entropy for the subsystem *query* is defined as: $S_{\text{query}} = \frac{1}{2}\text{Tr}[\log(\rho_{\text{query}})]$ with $\rho_{\text{query}} = \text{Tr}_{\text{key}}(|\Psi\rangle\langle\Psi|)$ being the reduced density matrix obtained similarly by tracing out the *key* subsystem from the state $|\Psi\rangle$. The Rényi entropy can also be efficiently calculated using classical shadows. It is typically simpler to compute both analytically and numerically compared to Von Neumann entropy (Wang et al., 2025).

A comparison of these entanglement measures was performed to choose the best measure. The experimental results are discussed in Section 6.

# 5 Swap test-based attention

Quantum kernel methods, which embed classical data into quantum states and compute their inner products (i.e., kernels), or directly evaluate overlaps between quantum input states, have emerged as promising

quantum machine learning (QML) techniques for quantum advantage (Liu et al., 2021; Jäger & Krems, 2023). Inspired by this approach, we introduce a swap test-based attention mechanism to serve as an additional baseline comparison. While similar to the entanglement-based attention model, the swap test-based approach replaces the Parameterized Quantum Circuit (PQC) and entanglement measurement with a swap-test circuit which computes the overlap between *query* and *key* quantum states. The architecture of this attention model is illustrated in Figure 1 and the swap test circuit is illustrated in Figure 5.

The SWAP test is a well-established quantum technique for evaluating the similarity between two pure $n$-qubit states $|\phi\rangle_{\text{query}}$ and $|\phi\rangle_{\text{key}}$. The initial system state is prepared as:

$$|\Psi\rangle = |\phi\rangle_{\text{query}} \otimes |\phi\rangle_{\text{key}} \otimes |0\rangle_C . \tag{5}$$

A Hadamard gate is then applied to the control qubit $C$, followed by a `controlled-SWAP` gate that exchanges *query* and *key* quantum states based on the state of the qubit $C$. The probability of measuring the control qubit in state $|1\rangle$ provides a direct measure of the overlap or similarity between $|\phi\rangle_{\text{query}}$ and $|\phi\rangle_{\text{key}}$.

## 6  Experiments

To assess the effectiveness of the proposed method, we employed various libraries to implement the hybrid approach. The simulation of quantum circuits was carried out using the `TensorCircuit` library (Zhang et al., 2023), while the `Equinox` library (Kidger & Garcia, 2021) was utilized to construct the Transformer architecture. Figure 1 displays the quantum-classical Transformer architecture, which builds upon the basic Transformer architecture featuring a single attention head. We apply attention layers in sequence. The combined query, key, and value vectors contribute to a total of $3 \times (\text{embed\_dim} \times \text{embed\_dim})$ trainable parameters. The number of trainable parameters within the PQC in the quantum attention corresponds to half the number of qubits utilized. The final linear layer contains $\text{embed\_dim} \times \text{n\_classes}$ parameters.

Quantum elements were incorporated into the attention layer, as detailed in the previous section. The query, key, and value vectors were computed from the input using a feed-forward network (without the bias term). These vectors were then mapped to quantum states using a quantum feature map and entangled using a Parameterized Quantum Circuit (PQC). The entanglement entropy between the states was assigned as the attention coefficient. We use only the CLS token output for classification to ensure that the performance of the model primarily depends on the attention layer.

**Evaluation metrics**  We report three performance metrics for the models: i) train accuracy, ii) test accuracy, and iii) test Nearest Exemplar Accuracy (NEA). We have added the NEA baseline, in order to test the effectiveness of the attention layer in extracting the relevant information for the target classification problem in isolation of the linear classification layer effect and capacity. For that we train the model while omitting the bias term from the classification layer, allowing us to treat the linear classification layer weights as prototypes of the corresponding classes. In the learned embedding space of the CLS token, we can compute another metric of classification accuracy based on the nearest class mean. We compute the mean feature vector of the training samples from each class and then assign to the test sample the label of the closest mean (exemplar) in terms of cosine similarity. We refer to this as Nearest Exemplar Accuracy (NEA). The NEA metric allows us to assess the quality of the extracted CLS token and the learned features in isolation of the optimized classification head.

We report only the interquartile mean (IQM) of the best accuracies across ten runs (with different seeds). This was used as an alternative to median and mean, as it corresponds to the mean score of the middle 50% of the runs combined across all tasks. This makes it more robust to outliers than the mean and a better indicator of overall performance than the median (Agarwal et al., 2021).

**Datasets**  We evaluate quantum attention model on the MC and RP datasets, previously used by Li et al. (2022) for evaluating QNLP models. MC contains 17 words and 100 sentences (70 train + 30 test) with 3 or 4 words each; RP has 115 words and 105 sentences (74 train + 31 test) with four words in each one. Each word is represented as a token and encoded as vectors using a word2vec model.

We also test the model performance on MNIST, FMNIST, and MNIST-1D datasets. MNIST-1D (Greydanus & Kobak, 2024) is a low-dimensional variant of MNIST that emphasizes learning non-linear representations for successful classification. Its small size and complexity make it a suitable dataset for testing quantum models on classical computers. It is reshaped and represented as four tokens. The MNIST and FMNIST dataset are resized to 12x12 using bilinear interpolation and are represented as 12 tokens.

Furthermore, all tokens are encoded into a vector of length 12.

**Hyperparameters and their tuning**   The primary goal of our experiments is to conduct a fair comparison of classical and quantum attention models. Thus, we did not engage in extensive hyperparameter tuning. We tried different batch sizes and learning rates and did not observe a significant qualitative difference. We chose to report the experiments obtained for a standard batch size of 64 and 128 for small and large datasets, respectively. We trained both quantum-, and classical models using Adam optimizer with learning rate of 0.01 and cosine decay schedule for 500 epochs. The train-test split for all datasets was 66.66/33.3. The embedding dimension; i.e., the length of the query and key vectors which are mapped to quantum states is 12. We used 4 data reuploading layers in PQC, which controls the expressiveness of the quantum model. We use 2 sequential attention layers for both quantum-, and classical Transformers.

**Compared models**   We compare our method against a classical Transformer utilizing scaled dot-product attention. All architectural components, except for the attention layer, are identical across both the classical and quantum models. This design ensures a fair and controlled comparison between the different attention mechanisms. The following approaches are included in the comparison:

1. **Rényi Entropy-based quantum attention (RE):** The proposed quantum attention mechanism employs Rényi entropy to quantify entanglement and measure similarity between query and key quantum states.

2. **Von Neumann Entropy-based quantum attention (VN):** A variant of the proposed quantum attention model that utilizes Von Neumann entropy as the entanglement measure for computing query-key similarity.

3. **Swap test-based quantum attention (ST):** A baseline quantum attention mechanism introduced in this work that leverages the swap test to compute the dot product between quantum states. This approach aligns with the intuition behind many existing quantum and classical attention mechanisms that use dot-product-based similarity.

4. **Quantum self-attention neural network (QSANN):** The attention model proposed by Li et al. (2022), which uses Gaussian Projected Quantum Self-Attention (GPQSA). In this method, quantum states are projected onto classical values via quantum measurements, followed by a Gaussian function applied to the resulting one-dimensional classical values.

5. **QSANN with CLS token:** A modified version of the QSANN model to align more closely with the classical Transformer architecture used in classification tasks. Instead of averaging the attention outputs across all tokens, a learnable `[CLS]` token is introduced, and only its output is used for classification.

6. **Quantum self-attention mechanism (QSAMo, QSAMb)**: A fully quantum implementation of the classical self-attention mechanism, focusing on quantum data encoding and the design of PQCs to compute attention vectors. QSAMo is an optimized model that removes repeated ending circuits in the QSAMb model.

The experiments carried out are described in the following subsections. In Appendix B and Appendix C, we report further experiment details, as well as additional experiments.

Table 1: **Comparison of various entanglement measures on text classification datasets.** The MC and RP datasets were used to compare the performance of entanglement-based attention models: Rényi entropy (RE) and Von Neumann entropy (VN). Additionally, we compare the swap test-based attention model (ST), the QSANN method, and the QSAMb and QSAMo models proposed by (Shi et al., 2023). While the swap test is not an entanglement measure, it is commonly used in the literature due to its ability to compute similarity (dot product) in a higher-dimensional space. We also report QSANN (CLS token), a modified version of QSANN that uses only the CLS token for classification. Performance is measured in terms of interquartile mean (IQM) and standard deviation. The results show that entanglement entropy outperforms all existing quantum approaches and classical attention but lags behind the swap test-based attention model, which is also proposed as a baseline in this work. This highlights the ability of entanglement to capture relationships between words, similar to dot-product-based attention models.

| Model | MC Train Acc. | MC Test Acc. | RP Train Acc. | RP Test Acc. |
|---|---|---|---|---|
| Classical | 100 | 100 | $99.73 \pm 1.22$ | $75.22 \pm 4.99$ |
| QSANN (CLS token) | 58.57 | 56.66 | 67.57 | 54.84 |
| QSANN (original) | 100.00 | 100.00 | 95.35 | $67.74 \pm 0$ |
| QSAMb | - | 100.00 | - | 72.58 |
| QSAMo | - | 100.00 | - | 74.19 |
| RE (this work) | 100 | 100 | 100 | $70.92 \pm 3.05$ |
| VN (this work) | 100 | 100 | 100 | $74.82 \pm 3.68$ |
| ST (proposed baseline) | 100 | 100 | 100 | $\mathbf{83.03 \pm 3.60}$ |

## 6.1 Performance of entanglement measures on text classification datasets

We evaluate the effectiveness of different entanglement measures on the MC and RP datasets. These two datasets are selected due to their small size and their adoption in prior studies on QSANN and QSAM (Shi et al., 2023). Table 1 summarizes the performance of various entanglement-based attention mechanisms and the swap test on these datasets. Our results show that the swap test-based attention model (proposed as a baseline in this work) outperforms all existing methods. The proposed entanglement entropy-based models also surpass previously introduced methods, including QSANN, QSAMb, and QSAMo by Shi et al. (2023), as well as the classical attention model.

Interestingly, the performance of QSANN drops considerably when only a CLS token is used for classification. The results for this modified version, denoted as QSANN (CLS token), are also reported in Table 1.

## 6.2 Performance of quantum attention model on text datasets under noisy conditions

In this section, we investigate the robustness of quantum attention mechanisms when subjected to real-istic quantum noise—an inherent challenge in quantum computing. Quantum noise arises from a variety of sources, including external thermal fluctuations, calibration imperfections, dephasing, and decoherence. These phenomena compromise the fidelity of quantum operations and, if a model is highly sensitive to such noise, limit its practical utility. Hence, evaluating model performance under noisy conditions is essential for assessing the real-world viability of quantum attention mechanisms.

To this end, we evaluate the proposed quantum attention models on the MC and RP text classification datasets under three types of noise: 1-qubit and 2-qubit depolarizing noise, and thermal relaxation noise. All simulations are conducted using the `TensorCircuit` library, which provides native support for realistic noise modeling. Depolarizing noise is simulated as the random application of Pauli gates (`X`, `Y`, `Z`), representing the loss of coherence due to environmental interactions. Thermal relaxation noise accounts for energy dissipation and loss of phase coherence, modeled via relaxation ($T_1$) and dephasing ($T_2$) processes.

To simulate realistic conditions, we adopt median calibration data from the IBM *Kyiv* quantum device (as of November 19, 2024). Specifically, we use $T_1 = 277.04\,\mu s$ and $T_2 = 117.71\,\mu s$. The single-qubit depolarizing noise probabilities are set to $p_{1x} = p_{1y} = p_{1z} = p_1 = 2.673 \times 10^{-4}$, while the two-qubit depolarizing probabilities are set to $p_{2x} = p_{2y} = p_{2z} = p_2 = 1.224 \times 10^{-2}$. The classical query and key vectors are encoded into quantum states using a dense encoding scheme.

Table 2: **Comparison of various entanglement measures on text classification datasets under quantum noise.** The MC and RP datasets were used to compare the performance of entanglement-based attention models: Rényi entropy (RE) and Von Neumann entropy (VN). Additionally, we compare the swap test-based attention model (ST) and the QSAMo model proposed by (Shi et al., 2023) under noise. The noise settings in the experiments simulate realistic conditions, with bit flip and depolarizing noise set using the median calibration data from the IBM Kyiv quantum device on November 19, 2024. Performance is measured in terms of Interquartile Mean (IQM) and standard deviation. The results demonstrate that entanglement entropy outperforms all existing quantum approaches, the swap test-based attention model, and classical attention under noise. These findings suggest that entanglement-based attention models may be well-suited for current noisy quantum devices.

| Model | MC Train Acc. | MC Test Acc. | RP Train Acc. | RP Test Acc. |
|---|---|---|---|---|
| Classical | 100 | 100 | 99.73 | $75.22 \pm 4.99$ |
| QSAMo (bit-flip) | - | 100.00 | - | 74.19 |
| QSAMo (Depolarizing) | - | 100.00 | - | 70.97 |
| QSAMo (Amplitude damping) | - | 100.00 | - | 67.74 |
| RE (this work) | 100 | 100 | 97.28 | $82.35 \pm 3.23$ |
| VN (this work) | 100 | 100 | 96.13 | $\mathbf{84.39 \pm 4.34}$ |
| ST (proposed baseline) | 100 | 100 | 96.51 | $83.50 \pm 3.40$ |

Previous work by Shi et al. (2023) evaluated the QSAMo model under various noise conditions—including bit flip, depolarizing, and amplitude damping noise—and reported significant performance degradation. However, in our experiments, we observe that the proposed entanglement-based quantum attention models exhibit strong robustness under noise. In fact, they outperform all baseline models, including the swap-test-based attention, even when the latter are evaluated in noise-free settings. This is particularly evident on the RP dataset, where entanglement-based attention models demonstrate superior generalization performance in the presence of noise.

These findings suggest that, in certain cases, quantum noise may act as a form of implicit regularization, enhancing the generalization ability of quantum models. We leave a deeper investigation of this phenomenon for future work. A detailed performance comparison of all models under noisy conditions is presented in Table 2.

## 6.3 Performance of attention models on image datasets

In this section, we extend our evaluation to image classification datasets to assess whether the effectiveness of quantum attention models generalizes beyond text-based tasks. Specifically, we include MNIST-1D, MNIST, and Fashion-MNIST (FMNIST) datasets. These datasets vary in complexity and size, providing a useful benchmark for analyzing model behavior under different data regimes. The performance results across different encoding schemes are presented in Tables 3 and 4.

We observe that entanglement-based attention mechanisms consistently outperform swap test-based models across all image datasets. This finding suggests that entanglement measures are more effective than quantum state similarity measures (as used in the swap test) for capturing structural relationships between image patches. These results support our central hypothesis: *Can entanglement model relationships between text tokens—and, by extension, image patches—in a manner analogous to its ability to represent correlations in many-body quantum systems?*

To further investigate whether quantum models offer an advantage over classical attention on small datasets, we systematically vary the dataset size by subsampling to 100, 1,000, and 10,000 samples. This enables a direct performance comparison between classical and quantum attention models at different data scales. As shown in Tables 3 and 4, quantum attention models—particularly those using entanglement-based measures—achieve higher test accuracy and lower Nearest Exemplar accuracy (NEA) than classical attention models on smaller datasets.

Table 3: **Performance of super-dense quantum attention models on various image datasets.** This table compares the performance of entanglement-based attention models—Rényi entropy (RE) and Von Neumann entropy (VN)—alongside the swap test-based attention model (ST) and classical dot-product-based attention across three datasets: MNIST, FMNIST, and MNIST-1D. For quantum attention models, super-dense encoding feature maps are utilized. Performance is measured in terms of Interquartile Mean (IQM) and standard deviation. Quantum entanglement-based attention models exhibit superior performance when fewer data points are available. This trend is evident in their improved results across all datasets when considering 100 and 1000 samples, as well as their strong performance on the RP and MC datasets in Table 1. Furthermore, entanglement entropy-based methods surpass swap test-based attention models, with Rényi entropy-based attention consistently outperforming Von Neumann entropy-based attention.

| Dataset | Model | Train Acc. | Test Acc. | Test NEA |
|---|---|---|---|---|
| MNIST-1D (100) | Classical | $100.00 \pm 0.00$ | $29.43 \pm 4.65$ | $28.50 \pm 4.31$ |
| | RE (this work) | $100.00 \pm 0.00$ | $31.20 \pm 4.20$ | $33.14 \pm 3.49$ |
| | VN (this work) | $100.00 \pm 0.00$ | $32.67 \pm 3.12$ | $\mathbf{36.00 \pm 2.86}$ |
| | ST (proposed baseline) | $100.00 \pm 0.00$ | $34.25 \pm 4.77$ | $32.40 \pm 4.74$ |
| MNIST-1D (1000) | Classical | $68.83 \pm 2.00$ | $43.92 \pm 2.11$ | $41.80 \pm 0.98$ |
| | RE (this work) | $68.80 \pm 2.44$ | $47.64 \pm 4.32$ | $\mathbf{42.90 \pm 1.63}$ |
| | VN (this work) | $64.95 \pm 2.65$ | $42.90 \pm 3.14$ | $41.95 \pm 1.31$ |
| | ST (proposed baseline) | $56.82 \pm 3.28$ | $43.80 \pm 2.67$ | $41.30 \pm 1.67$ |
| MNIST-1D (10000) | Classical | $66.65 \pm 3.63$ | $65.18 \pm 3.67$ | $\mathbf{59.90 \pm 3.05}$ |
| | RE (this work) | $57.00 \pm 4.18$ | $55.09 \pm 4.29$ | $\mathbf{53.07 \pm 4.21}$ |
| | VN (this work) | $48.95 \pm 4.74$ | $47.26 \pm 4.48$ | $44.68 \pm 3.73$ |
| | ST (proposed baseline) | $46.78 \pm 4.69$ | $45.46 \pm 4.84$ | $43.14 \pm 4.72$ |
| MNIST (100) | Classical | $100.00 \pm 0.00$ | $57.00 \pm 3.82$ | $55.50 \pm 3.28$ |
| | RE (this work) | $100.00 \pm 0.00$ | $66.00 \pm 2.44$ | $\mathbf{65.00 \pm 3.25}$ |
| | VN (this work) | $100.00 \pm 0.00$ | $66.40 \pm 4.33$ | $63.60 \pm 5.10$ |
| | ST (proposed baseline) | $100.00 \pm 0.00$ | $68.33 \pm 4.40$ | $\mathbf{65.20 \pm 5.57}$ |
| MNIST (1000) | Classical | $99.50 \pm 0.35$ | $85.30 \pm 1.07$ | $\mathbf{82.05 \pm 1.85}$ |
| | RE (this work) | $97.38 \pm 0.85$ | $81.80 \pm 1.27$ | $\mathbf{76.50 \pm 1.82}$ |
| | VN (this work) | $94.72 \pm 1.47$ | $81.48 \pm 0.90$ | $73.76 \pm 1.35$ |
| | ST (proposed baseline) | $90.76 \pm 0.95$ | $80.60 \pm 1.12$ | $73.96 \pm 1.77$ |
| MNIST | Classical | $93.26 \pm 0.36$ | $92.66 \pm 0.44$ | $\mathbf{88.58 \pm 1.36}$ |
| | RE (this work) | $91.37 \pm 0.46$ | $90.67 \pm 0.45$ | $82.85 \pm 1.26$ |
| | VN (this work) | $90.55 \pm 0.31$ | $90.06 \pm 0.38$ | $\mathbf{85.35 \pm 1.39}$ |
| | ST (proposed baseline) | $84.95 \pm 0.48$ | $84.69 \pm 0.57$ | $76.08 \pm 1.34$ |
| FMNIST (100) | Classical | $100.00 \pm 0.00$ | $67.33 \pm 4.20$ | $\mathbf{69.20 \pm 4.49}$ |
| | RE (this work) | $100.00 \pm 0.00$ | $71.50 \pm 3.26$ | $\mathbf{66.00 \pm 3.49}$ |
| | VN (this work) | $100.00 \pm 0.00$ | $71.00 \pm 3.58$ | $63.00 \pm 4.00$ |
| | ST (proposed baseline) | $100.00 \pm 0.00$ | $71.67 \pm 2.65$ | $66.00 \pm 3.90$ |
| FMNIST (1000) | Classical | $97.15 \pm 0.67$ | $76.12 \pm 1.03$ | $\mathbf{75.25 \pm 0.76}$ |
| | RE (this work) | $94.43 \pm 0.96$ | $75.60 \pm 1.00$ | $\mathbf{72.70 \pm 0.98}$ |
| | VN (this work) | $91.34 \pm 1.32$ | $74.85 \pm 0.71$ | $71.40 \pm 0.86$ |
| | ST (proposed baseline) | $89.92 \pm 1.10$ | $73.84 \pm 1.11$ | $70.28 \pm 0.77$ |
| FMNIST | Classical | $84.68 \pm 0.30$ | $83.85 \pm 0.27$ | $\mathbf{81.20 \pm 0.58}$ |
| | RE (this work) | $83.18 \pm 0.31$ | $82.56 \pm 0.31$ | $\mathbf{78.18 \pm 0.87}$ |
| | VN (this work) | $81.27 \pm 0.41$ | $80.57 \pm 0.38$ | $75.68 \pm 0.58$ |
| | ST (proposed baseline) | $81.38 \pm 0.17$ | $80.73 \pm 0.18$ | $76.21 \pm 0.77$ |

Table 4: **Performance of dense quantum attention models on various image datasets.** This table compares the performance of entanglement-based attention models—Rényi entropy (RE) and Von Neumann entropy (VN)—alongside the swap test-based attention model (ST) and classical dot-product-based attention across three datasets: MNIST, FMNIST, and MNIST-1D. Performance is measured in terms of interquartile mean (IQM) and standard deviation. For quantum attention models, dense encoding feature maps are employed. The results align with observations from super-dense encoding, showing that quantum entanglement-based attention models perform better when fewer data points are available. This trend is particularly evident in their improved results across all datasets when considering 100 and 1000 samples, as well as their superior performance on the RP and MC datasets in Table 1. Furthermore, entanglement entropy-based methods outperform swap test-based attention models, with Rényi entropy-based attention consistently achieving better results than Von Neumann entropy-based attention.

| Dataset | Model | Train Acc. | Test Acc. | Test NEA |
|---|---|---|---|---|
| MNIST-1D (100) | Classical | $100.00 \pm 0.00$ | $29.43 \pm 4.65$ | $28.50 \pm 4.31$ |
| | RE (this work) | $100.00 \pm 0.00$ | $33.00 \pm 5.17$ | $\mathbf{34.00 \pm 3.98}$ |
| | VN (this work) | $100.00 \pm 0.00$ | $31.20 \pm 4.72$ | $29.33 \pm 4.40\ 3$ |
| | ST (proposed baseline) | $100.00 \pm 0.00$ | $32.00 \pm 4.40$ | $29.50 \pm 4.56$ |
| MNIST-1D (1000) | Classical | $68.83 \pm 2.00$ | $43.92 \pm 2.11$ | $41.80 \pm 0.98$ |
| | RE (this work) | $71.53 \pm 2.09$ | $44.00 \pm 3.00$ | $\mathbf{43.05 \pm 0.93}$ |
| | VN (this work) | $65.10 \pm 2.96$ | $42.95 \pm 3.49$ | $41.90 \pm 1.38$ |
| | ST (proposed baseline) | $54.80 \pm 2.20$ | $43.70 \pm 2.01$ | $41.64 \pm 1.71$ |
| MNIST-1D (10000) | Classical | $66.65 \pm 3.63$ | $65.18 \pm 3.67$ | $\mathbf{59.90 \pm 3.05}$ |
| | RE (this work) | $59.63 \pm 4.25$ | $57.15 \pm 4.42$ | $54.47 \pm 3.75$ |
| | VN (this work) | $60.00 \pm 4.99$ | $57.78 \pm 5.06$ | $\mathbf{54.60 \pm 4.22}$ |
| | ST (proposed baseline) | $47.70 \pm 2.89$ | $46.03 \pm 3.02$ | $43.19 \pm 3.45$ |
| MNIST (100) | Classical | $100.00 \pm 0.00$ | $57.00 \pm 3.82$ | $55.50 \pm 3.28$ |
| | RE (this work) | $100.00 \pm 0.00$ | $63.14 \pm 4.04$ | $61.00 \pm 5.33$ |
| | VN (this work) | $100.00 \pm 0.00$ | $59.00 \pm 4.57$ | $56.86 \pm 4.63$ |
| | ST (proposed baseline) | $100.00 \pm 0.00$ | $67.50 \pm 4.75$ | $\mathbf{64.50 \pm 5.93}$ |
| MNIST (1000) | Classical | $99.50 \pm 0.35$ | $85.30 \pm 1.07$ | $\mathbf{82.05 \pm 1.85}$ |
| | RE (this work) | $99.00 \pm 0.52$ | $83.05 \pm 1.11$ | $\mathbf{78.05 \pm 2.03}$ |
| | VN (this work) | $97.22 \pm 0.71$ | $80.10 \pm 1.44$ | $74.96 \pm 1.58$ |
| | ST (proposed baseline) | $91.15 \pm 0.68$ | $80.80 \pm 1.00$ | $73.20 \pm 1.25$ |
| MNIST | Classical | $93.26 \pm 0.36$ | $92.66 \pm 0.44$ | $\mathbf{88.58 \pm 1.36}$ |
| | RE (this work) | $91.14 \pm 0.24$ | $90.58 \pm 0.27$ | $\mathbf{85.92 \pm 0.88}$ |
| | VN (this work) | $90.60 \pm 0.18$ | $90.17 \pm 0.22$ | $83.64 \pm 1.42$ |
| | ST (proposed baseline) | $86.22 \pm 0.63$ | $85.92 \pm 0.63$ | $77.73 \pm 1.43$ |
| FMNIST (100) | Classical | $100.00 \pm 0.00$ | $67.33 \pm 4.20$ | $\mathbf{69.20 \pm 4.49}$ |
| | RE (this work) | $100.00 \pm 0.00$ | $69.00 \pm 2.89$ | $\mathbf{65.60 \pm 3.69}$ |
| | VN (this work) | $100.00 \pm 0.00$ | $70.44 \pm 2.19$ | $64.00 \pm 2.09$ |
| | ST (proposed baseline) | $100.00 \pm 0.00$ | $71.60 \pm 2.56$ | $65.20 \pm 3.25$ |
| FMNIST (1000) | Classical | $97.15 \pm 0.67$ | $76.12 \pm 1.03$ | $\mathbf{75.25 \pm 0.76}$ |
| | RE (this work) | $95.30 \pm 1.05$ | $75.65 \pm 1.04$ | $\mathbf{73.20 \pm 1.37}$ |
| | VN (this work) | $95.48 \pm 0.79$ | $75.20 \pm 0.99$ | $72.30 \pm 0.89$ |
| | ST (proposed baseline) | $88.97 \pm 0.88$ | $73.80 \pm 1.08$ | $70.45 \pm 0.81$ |
| FMNIST | Classical | $84.68 \pm 0.30$ | $83.85 \pm 0.27$ | $\mathbf{81.20 \pm 0.58}$ |
| | RE (this work) | $83.20 \pm 0.27$ | $82.52 \pm 0.30$ | $\mathbf{78.52 \pm 0.61}$ |
| | VN (this work) | $82.78 \pm 1.06$ | $82.00 \pm 1.00$ | $77.72 \pm 1.28$ |
| | ST (proposed baseline) | $81.64 \pm 0.20$ | $80.94 \pm 0.19$ | $76.23 \pm 0.88$ |

However, classical attention begins to outperform quantum models as the dataset size increases. Despite this, entanglement-based attention remains competitive and consistently outperforms the swap test-based baseline. For example, on the MNIST-1D dataset with 10,000 samples, both the classical and entanglement-based models achieve similar test NEA values, indicating that entanglement-based models scale well. On the full MNIST and FMNIST datasets, the test accuracy of entanglement-based models closely approaches that of the classical Transformer, while swap test-based models lag behind in all scenarios.

Table 5: **Performance of super-dense quantum attention models on various small image datasets under quantum noise.** This table compares the performance of entanglement-based attention models—Rényi entropy (RE) and Von Neumann entropy (VN)—alongside the swap test-based attention model (ST) and classical dot-product-based attention across three datasets: MNIST, FMNIST, and MNIST-1D. Performance is measured in terms of interquartile mean (IQM) and standard deviation. To optimize qubit usage, quantum attention models employ super-dense encoding feature maps. The results are consistent with noisy simulations on RP datasets, showing a reduction in the train-test accuracy gap under noise. However, noise negatively impacts overall model performance. Quantum attention models perform comparably or better on smaller dataset sizes, particularly for MNIST and MNIST-1D. Additionally, entanglement entropy-based attention consistently outperforms the swap test-based approach.

| Dataset | Model | Train Acc. | Test Acc. | Test NEA |
|---|---|---|---|---|
| MNIST-1D (100) | Classical | $100.00 \pm 0.00$ | $29.43 \pm 4.65$ | $\mathbf{28.50 \pm 4.31}$ |
| | RE (this work) | $74.00 \pm 4.36$ | $30.00 \pm 5.02$ | $26.50 \pm 4.49$ |
| | VN (this work) | $74.75 \pm 3.58$ | $31.00 \pm 5.60$ | $\mathbf{27.00 \pm 4.31}$ |
| | ST (proposed baseline) | $74.00 \pm 5.02$ | $30.00 \pm 4.21$ | $25.71 \pm 4.31$ |
| MNIST-1D (1000) | Classical | $68.83 \pm 2.00$ | $43.92 \pm 2.11$ | $\mathbf{41.80 \pm 0.98}$ |
| | RE (this work) | $54.53 \pm 1.81$ | $44.40 \pm 2.31$ | $\mathbf{41.60 \pm 1.78}$ |
| | VN (this work) | $54.55 \pm 1.65$ | $44.55 \pm 2.36$ | $41.55 \pm 1.86$ |
| | ST (proposed baseline) | $54.40 \pm 1.65$ | $43.95 \pm 1.61$ | $40.95 \pm 1.56$ |
| MNIST (100) | Classical | $100.00 \pm 0.00$ | $57.00 \pm 3.82$ | $55.50 \pm 3.28$ |
| | RE (this work) | $100.00 \pm 0.00$ | $63.50 \pm 4.24$ | $\mathbf{62.80 \pm 4.85}$ |
| | VN (this work) | $100.00 \pm 0.00$ | $66.40 \pm 4.33$ | $63.60 \pm 5.10$ |
| | ST (proposed baseline) | $100.00 \pm 0.00$ | $63.20 \pm 4.10$ | $61.60 \pm 4.90$ |
| MNIST (1000) | Classical | $99.50 \pm 0.35$ | $85.30 \pm 1.07$ | $\mathbf{82.05 \pm 1.85}$ |
| | RE (this work) | $90.60 \pm 0.54$ | $80.37 \pm 1.03$ | $73.20 \pm 1.56$ |
| | VN (this work) | $90.42 \pm 0.54$ | $80.55 \pm 1.03$ | $73.40 \pm 1.57$ |
| | ST (proposed baseline) | $90.40 \pm 0.52$ | $80.45 \pm 0.81$ | $73.48 \pm 1.36$ |
| FMNIST (100) | Classical | $100.00 \pm 0.00$ | $67.33 \pm 4.20$ | $\mathbf{69.20 \pm 4.49}$ |
| | RE (this work) | $100.00 \pm 0.00$ | $68.00 \pm 2.15$ | $62.40 \pm 3.25$ |
| | VN (this work) | $100.00 \pm 0.00$ | $66.86 \pm 2.27$ | $62.40 \pm 3.25$ |
| | ST (proposed baseline) | $100.00 \pm 0.00$ | $67.20 \pm 2.50$ | $62.86 \pm 2.94$ |
| FMNIST (1000) | Classical | $97.15 \pm 0.67$ | $76.12 \pm 1.03$ | $\mathbf{75.25 \pm 0.76}$ |
| | RE (this work) | $75.72 \pm 0.47$ | $68.50 \pm 0.57$ | $64.88 \pm 1.13$ |
| | VN (this work) | $75.95 \pm 0.41$ | $68.40 \pm 0.62$ | $64.88 \pm 1.04$ |
| | ST (proposed baseline) | $75.12 \pm 0.70$ | $68.26 \pm 0.46$ | $63.85 \pm 0.95$ |

## 6.4 Performance of attention models on image datasets under quantum noise

Given the high computational cost of simulating noise in large-scale quantum circuits, we restrict our noise evaluation to smaller versions of the image datasets (100 and 1,000 samples) using the super-dense encoding scheme. The goal is to understand the impact of realistic quantum noise on model performance. The results are presented in Table 5.

We observe that the presence of noise slightly degrades model performance on small datasets. However, it also reduces the train-test accuracy gap, indicating improved generalization. This suggests that quantum noise can act as a form of regularization in hybrid quantum-classical models. Under noisy conditions,

classical attention models outperform quantum models. Nevertheless, entanglement-based and swap test-based attention mechanisms perform comparably to each other.

### 6.5 Performance of quantum attention models with IQP encoding

We evaluate the performance of quantum attention mechanisms using Instantaneous Quantum Polynomial-time (IQP) encoding to map classical vectors into quantum states. IQP encoding is often considered one of the most inherently "quantum" data encoding techniques and is believed to have the potential for offering quantum computational advantages. However, the circuit complexity associated with IQP encoding introduces significant simulation overhead, especially for systems with a higher qubit count.

To address this limitation, we conduct our evaluation on the MNIST-1D dataset, where each token is encoded as a vector of size three. This setup allows us to represent both the query and key vectors using a total of six qubits, thereby keeping the simulation computationally tractable.

Under noiseless conditions, the test accuracies achieved by the classical attention model, the entanglement-based quantum attention model, and the swap test-based quantum attention model are 45.55%, 42.43%, and 42.40%, respectively. Notably, the results suggest that increased quantum circuit complexity—such as that introduced by IQP encoding—does not necessarily translate to improved performance. This highlights the challenge of identifying encoding techniques that effectively leverage the strengths of quantum models in practical machine learning tasks.

Due to the substantial simulation time required for IQP circuits, we do not evaluate this encoding strategy on larger datasets or with higher-dimensional token vectors.

## 7 Discussion

In this work, we proposed entanglement-based quantum attention models that utilized entanglement entropy to evaluate the similarity between the query and the key vectors. Although our results on large-scale classical datasets did not show a clear performance advantage over classical attention, the proposed quantum attention models demonstrated notable benefits in specific settings.

In particular, quantum attention exhibited improved generalization in smaller datasets, as observed in the MC and RP text classification tasks and in reduced versions of MNIST and MNIST-1D. These results suggest that quantum attention may be particularly well-suited for applications with limited data availability, such as those in the medical or scientific domains. However, determining the precise dataset size or data regime in which classical attention begins to consistently outperform quantum models remains an open research question. Addressing this could help identify the contexts in which quantum attention offers the greatest practical benefit.

We also evaluated the robustness of the models under realistic noisy conditions by simulating noise using parameters derived from IBM quantum hardware. The results showed that, while the noise slightly degraded overall performance, it reduced the generalization gap, indicating a potential regularization effect. Interestingly, on the RP dataset, noisy quantum attention models outperformed their noiseless counterparts, suggesting that in certain cases, noise enhanced model generalization and prevented overfitting. This phenomenon merits further investigation.

Our study explored three types of feature maps for encoding classical data into quantum states: (i) super-dense encoding, (ii) dense encoding, and (iii) Instantaneous Quantum Polynomial-time (IQP) encoding. Across experiments, quantum attention generally performed better on smaller datasets, regardless of the encoding scheme. However, we did not observe a consistent trend favoring one encoding method over another. In particular, IQP encoding, despite its higher circuit complexity, did not yield superior performance, underscoring the difficulty of identifying encoding strategies that could effectively exploit the potential benefits arising from the quantum nature of the encoding.

Overall, our findings indicate that entanglement entropy provides a meaningful similarity measure for quantum attention mechanisms, akin to how entanglement quantifies correlations in many-body quantum systems.

To our knowledge, this was the first demonstration of using entanglement-based metrics within an attention mechanism for classical machine learning tasks. Furthermore, we found that entanglement-based attention consistently outperforms the swap test, a widely adopted technique in quantum machine learning, for this purpose.

## 8 Limitations and future work

To fully harness the potential of quantum attention, future research should focus on evaluating the proposed model on larger datasets. Understanding its behavior with varying qubit counts is essential for assessing scalability and identifying datasets where the model exhibits an inductive bias, paving the way for practical applications.

In this work, we employed Quantum Feature Maps (QFMs) to encode classical vectors into quantum states. Future research could explore parameterized QFMs, which enhance expressivity and may lead to improved performance. Furthermore, all experiments were conducted on a classical simulator, limiting both scalability and dataset size. Evaluating the model on quantum hardware with a larger qubit count is a crucial next step.

Moreover, we assumed *static noise* in our quantum circuits, which means that the noise characteristics remained unchanged throughout the learning process. However, real quantum systems experience *noise drift*, where noise varies over time. Accounting for this effect is beyond the scope of this study, but it presents an interesting direction for future research.

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

## A  Quantum circuits of quantum feature maps and swap test

The circuit schematics of different quantum encoding strategies employed in this work. Refer to Figure 3 (super-dense), Figure 2 (dense), Figure 4 (IQP) and Figure 5 (swap test).

## B  Attention heatmaps

We present the attention coefficients computed by the first layer of quantum and classical attention mechanisms on the RP dataset. This provides insight into the features that the models prioritize. The attention heatmaps are shown in Figure 6. The attention coefficients demonstrate that the quantum attention layers can focus on significant features in the data, leveraging the entanglement entropy between the query and key states.

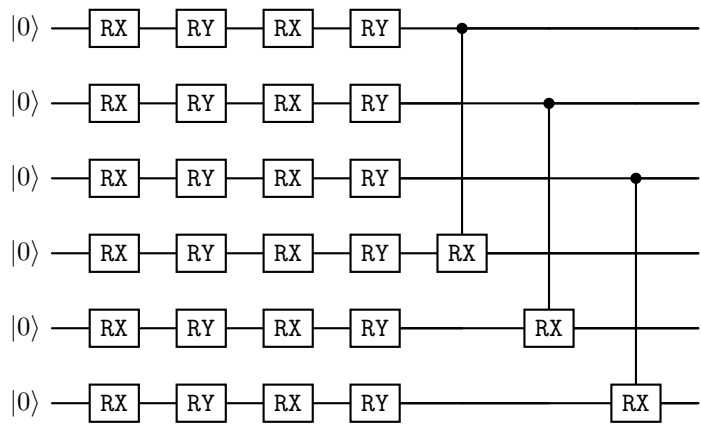

Figure 2: **Super-dense angle encoding circuit.** Encoding query and key vectors (of length 12) into the parameters of `RX` and `RY` gates using 6 qubits. This is followed by a PQC made of `controlled-RX` gates.

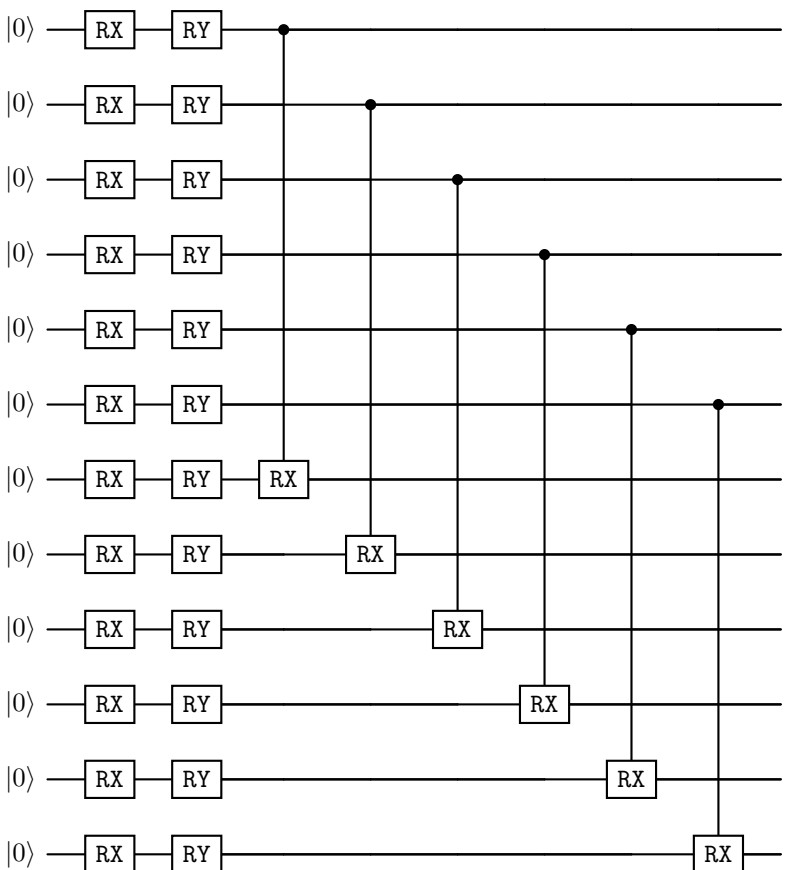

Figure 3: **Dense angle encoding circuit.** Encoding query and key vectors (of length 12) into the parameters of `RX` and `RY` gates using 12 qubits. This is followed by a PQC made of `controlled-RX` gates.

Additionally, it can be observed that these coefficients exhibit distinct patterns for each class. This characteristic differentiates the attention layers from a simple multi-layer perceptron (MLP) layer, which assigns uniform coefficients to all features.

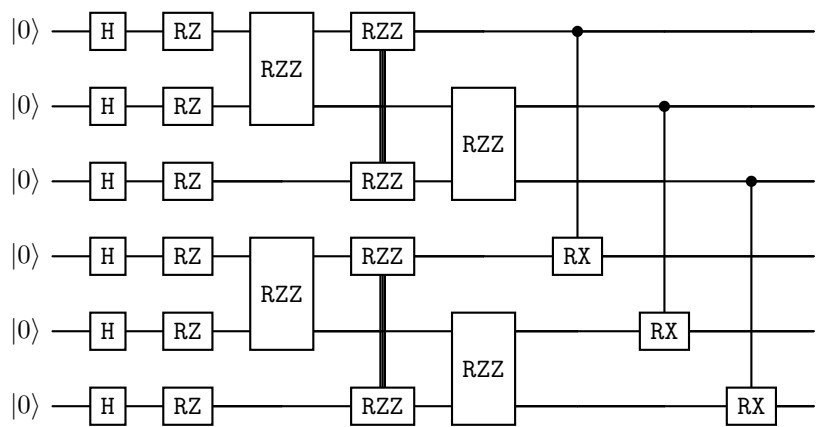

Figure 4: **IQP encoding circuit.** Encoding query and key vectors (of length 3) into the parameters of `RZ` and `RZZ` gates using 6 qubits. This is followed by a PQC made of `controlled-RX` gates.

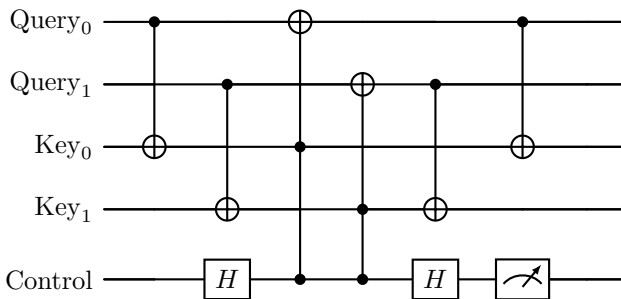

Figure 5: **Swap test circuit.** A dot product similarity is computed in the quantum embedded space using the swap test circuit.

## C Comparison with MLP

We compare the performance of the quantum attention model with a Multi-Layer Perceptron (MLP) on the RP and MC datasets. The MLP consists of three hidden layers and one output layer. Each hidden layer employs weight matrices of size $48 \times 48$, and the output layer has a weight matrix of size $48 \times 2$, resulting in approximately $7,000$ trainable parameters—substantially more than the quantum attention-based models. Note that, as the RP dataset contains 4 tokens, each of length 12, the dimensions of the weight matrices are $48 \times 48$ and $48 \times 2$.

For the MC dataset, the MLP achieves 100% accuracy on the test set with ease (see Figure 8). However, for the RP dataset, the MLP struggles to generalize and exhibits significant overfitting (see Figure 7). This experiment highlights that the quantum attention models do not overfit and can generalize well compared to an MLP. It also indicates that entanglement-based attention models are able to select important features based on entanglement entropy and do not degenerate into an MLP.

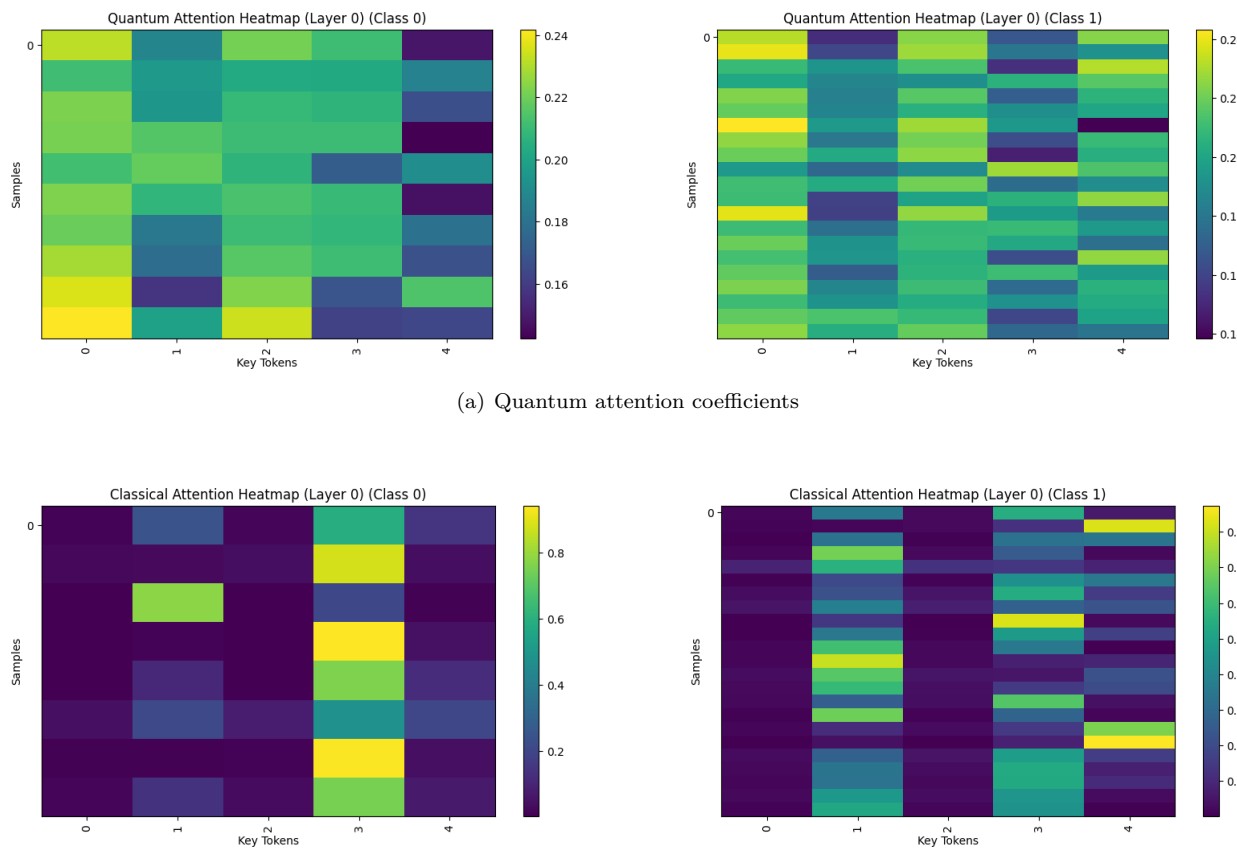

(a) Quantum attention coefficients

(b) Classical attention coefficients

Figure 6: **Quantum and classical attention heatmaps.** The heatmaps of the attention coefficients (using Renyi entropy) for the CLS token, calculated with respect to all other tokens in the RP dataset, are shown here. The coefficients are derived after applying the softmax activation. These heatmaps highlight the attention or importance given by the attention layers to each token while computing the output. Each row corresponds to a sample from a particular class. The plot on the left (right) displays the attention coefficients for all samples belonging to class 0 (class 1) with respect to the CLS token. The samples are grouped according to their predicted class. Both the quantum and classical attention models demonstrate the ability to assign importance to specific tokens, capturing distinct attention patterns for each class. This confirms that both models successfully learn the attention mechanism, varying the level of importance based on class-specific features.

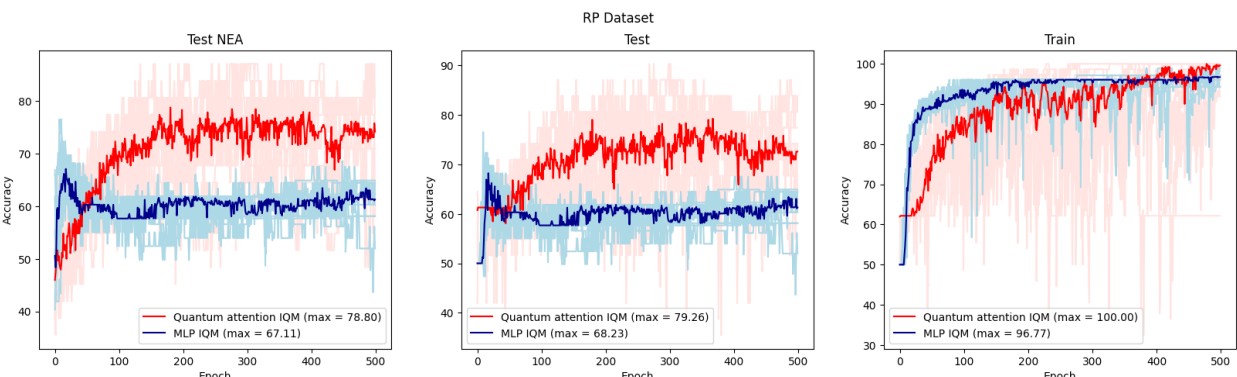

Figure 7: Comparison of quantum attention with MLP on RP dataset.

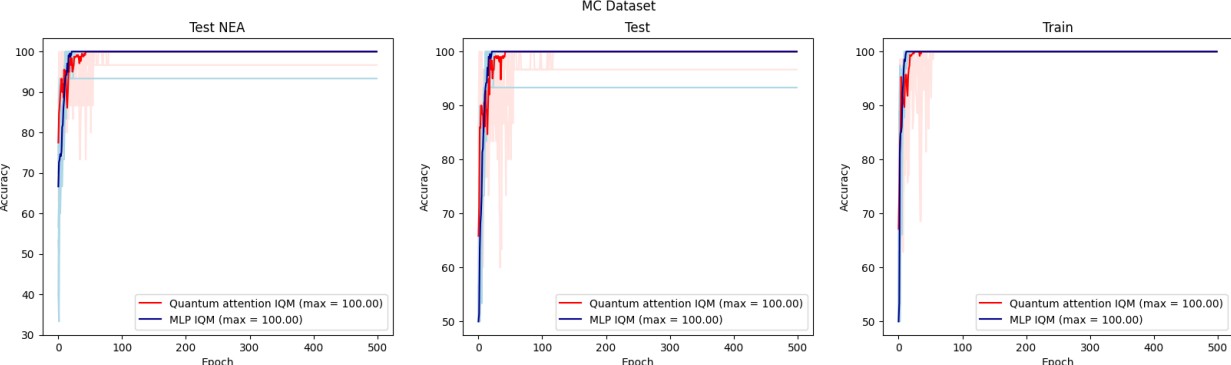

Figure 8: Comparison of quantum attention with MLP on MC dataset.

