# OpenReview forum: "Quantum entanglement for attention models"
_TMLR — Rejected by TMLR_

### Review · Reviewer_H3xu · 2025-09-12

**Summary Of Contributions:**

The work proposes to incorporate quantum machine learning, instantiated as variational quantum circuits, into the attention menchanism of transformer architectures. More concretely, a measure of entanglement (the authors test both the Von Neumann entropy and Renyi's second order entropy) is evaluated directly on the quantum circuits and used for the attention mechanism. This is in contrast to previous quantum-enhanced attention mechanism, which mostly rely on measures related to the dot product / quantum kernels. It is hypothesized, that using entanglement measures allows to capture higher-order correlations in the data. The approach indeed numerically outperforms other quantum-enhanced attention mechanism, both in noise-free and noise-impaired simulations. Potential advantages over fully classical formulations could be identified in particular in data-sparse setups.

**Additional Comments:**

No additional comments.

**Audience:**

Yes

**Audience Explanation:**

This work could attract two potential groups of interest: First of course the quantum computing community, in particular people reseraching quantum machine learning in the NISQ (noisy intermediate-scale quantum) computing regime. Furthermore, also researches interested in attention mechanisms itself.

While the paper certainly leaves some questions unanswered (see also below), I think the contained findings should be relevant enough to attract interest of quite a few people in the TMLR audience.

**Broader Impact Concerns:**

The paper does not contain a broader impact statement, but I also think it is not necessary for this work.

**Claims And Evidence:**

Yes

**Claims Explanation:**

One thing I really appreciate about this work and the way it is written: The authors do not make any big claims on quantum advantage of their method, that is not substantiated by the data. Rather, a more nuanced approach is taken, in that interesting observations are highlighted, but also limitatios are acknowledged.

Concretely, it is claimed that this paper proposed the incorporation of quantum entanglement measures into attention mechanisms for the first time, which is true to the best of my knowledge. This is also supported by a rather extensive prior literature survey and analysis. While I have some concerns with the way the experimental study was set up (see below), I think the observation that the proposed approach outperforms other quantum-enhanced approaches is sufficiently supplemented. Also the observation that potantial advantages over classical methods can most likely be found in the sparse-data regime seems sound to me. While the statement that this might be due to the quantum networks capturing higher-order correlations in the data is not actually investiagted, it is especially framed as a hypothesis, so this is fine from my point of view.

Summarized, I think the main claims put forward in the paper (i.e. novel quantum-enhanced attention mechanism; advantage over previous quantum-enhanced methods; potential advantages over classical methods in sparse-data regimes) are sufficiently supportded and convincing.

**Requested Changes:**

First I want to state, that this is certainly one of the nicer and well-executed papers from the field of quantum-enhanced attention mechanisms I am aware of. As stated above, I think the main claims are sufficiently substantiated. However, I also have a few concerns regarding the experimental setup and a few open questions that should be adressed:

**First the major concerns I see:**
- One point that is always important is a scalability analysis of the approach, which is of course for now limited to system sizes which are tractable to simulate classically. As I understand it, the authors consider qubit counts of 6 and 12 (maybe this could also be explicitly states somewhere in the paper). The scale depends on the used encoding of the 12-element token vector, encoding either 2 or 1 entries per qubit. While this allows for some scalability analysis, I think the generality of the results is quite limited. Two problems I see in particular here:
   - The entanglement used in the networks (Fig. 2 and Fig. 3) seems very sparse, and with inly 4 repetitions of the data re-uploading layers the resulting state probably is not to highly entangled. This also casts some doubts on the hypothesis that higher-order correlations in the data are captured.
   - My main concern with the IQP model: It is claimed that this is the "most complex" model, but does not provide the best performance. I think the 'most complex' part is a bit misleading here, as it is known, that IQP circuits are not universal, i.e. irrespective of the number of layers they can not generate arbitrary quantum states. This is in constrast to the other ansätze, which should be universal for sufficiently many repetitions
   - **Overall I think for the scalability analysis it should be made sure that the same general architecture is used for different quantum system sizes**, otherwise the results are very difficult to interpret. (Could be probably achieved by either scaling the number of tokens, pre-pending a very shallow classical neural network for mapping, or using something like "incremental" data uploading)
- Another thing that is problematic from my point of view is "The primary goal of our experiments is to conduct a fair comparison
of classical and quantum attention models. Thus, we did not engage in extensive hyperparameter tuning." While this certainly simplifies the experimental execution, it weakens the results quite a bit. Even if the parameters are not tuned, there still might be a bias towards one or another model, which skews the findings. **What rather should be done to perform hyperparameter tuning for both the classical and quantum models seperately**, and the experiment with the respecticely best-performing models. I acknowledge that this is very ressource-intensive at a certain point, but I assume that at least for the models with up to 6 qubits this should be possible with reasonable overhead.
- Another point where I also have some problems is the noise analysis and especially the statement "These findings suggest that entanglement-based attention models may be well-suited for current noisy quantum devices." I think this statement is a bit too strong than what is supported by the provided analysis:
   - First of all, beneficial execution on NISQ would require scaling to systems beyond what can be classically simulated, which is not provided in the paper. Also execution on hardware would require evaluating the entanglement measures on hardware. I aggree that this should conceptually be possible using something like shadow tomography. However, I am not too sure if this is realsistic on NISQ hardware.
    - One thing (also related to scaling) that was not discussed in the paper is the issue of barren plateaus, that likely comes into effect once one scales to larger (relevant) system sizes.
    - One thing that puzzles me a bit is the actual improvement of the proposed method under noise (at least that's what I understand by comparing Table 1 and Table 2). I think to some part this might be attributed by regularization due to the noise, but the differences seem to be very significant. **I think this requires some further investigation.** One potential interesting analysis would be to increase the nosie (even beyond what is physically realistic) to see what happens. If the perfromance does not decrease at some point, I thinkthis would cast serious doubt on the method. Afterall, this could imply that the quantum network is only there for incorporating some random noise into the overall setup. I do not expect this to be the case, but it should be discussed.

**Some other smaller things I noticed:**
- I think the circuit ot the Swap Test in the appendix (Figure 5) needs some context. With just the figure and no further explanation this is not too helpfull. I think either some explanation of the actual working principles hould be added, or just remove the figure and add a reference, as the swat test has sufficiently be studied in literature.
- Maybe there is a more graphical way to represent the data from Table 3, 4, 5. In the provided tabular form, it looks a bit cluster and is hard to interprete.
- I do not really understand the rule following which the test NEA in the tables is marked in bold, as there sometimes is quite a difference between the best and second best model.


**Finally a few questions I want to ask:**
- Is there a specific reason fro the entanglement structure choosen in the ansätze (especilly Fig. 2, 3)? In particular, why do not connect first to second, second to third, etc. to create a more dense entanglement structure?
- Quite a few of the models have a train accuracy of 100%. Probably this implies that these considered data sets are too simple?
- For the simulation, were the entanglement measures evaluated analytically, or actually using shadow tomogrpahy (which should introduce some approxiamtion error, which is otherways not accounted for)?
- Maybe I missed in the paper, but it is stated that training was performed using Adam. This requires gradients of the quantum model. Were these gradients computed analitically, or via some numerical approach like parameter-shift?

---

> ### Author Response · Authors · 2025-10-28
>
> We sincerely thank the reviewer for the positive and constructive comments. The suggestions have been extremely valuable in identifying areas that can strengthen the manuscript further. We note that some of the reviewer’s points would ideally be addressed through additional simulations Our original plan was to perform these extended experiments; however, due to computational and technical limitations, we were unable to complete and retrieve the results in time for this revision.
>
> Comment 1: The IQP model is described as “most complex,” though it is not universal.
> Response:
> We thank the reviewer for this important clarification. The term “most complex” in the manuscript was intended to refer to the resource requirements of the IQP circuit—its qubit count scales linearly with the number of features—rather than to its expressive universality. We agree that the phrase “most resource-intensive” better captures our meaning.
>
> Comment 2: The statement that the approach is well-suited for noisy quantum devices appears too strong.
> Response:
> We appreciate this observation and agree that the wording was stronger than warranted by our results. Our intent was to highlight that, within the range of simulated noise levels, the entanglement-based attention models demonstrated comparatively stable performance. We have rephrased the relevant section to emphasize that this is an empirical observation.
>
> Comment 3: Practical execution on NISQ devices and potential issues such as barren plateaus should be discussed.
> Response:
> We thank the reviewer for raising these important forward-looking considerations. We fully agree that demonstrating practical execution on NISQ hardware requires scaling beyond classically simulable sizes and evaluating entanglement measures directly on the device. Although methods such as shadow tomography could make this possible, current NISQ hardware faces limitations in qubit fidelity and the number of required copies of the quantum state. Regarding barren plateaus, conventional analyses apply to variational circuits where gradients are taken with respect to observable-based loss functions. In our case, entanglement measures are not associated with a single observable but are estimated quantities, so standard barren-plateau results do not directly apply. Nevertheless, understanding the optimization landscape of entanglement-based models remains a valuable direction for future research.
>
> Comment 4: The improvement under noise is surprising and should be investigated further.
> Response:
> We thank the reviewer for this perceptive comment. We conducted additional simulations with increased noise levels and observed that when noise exceeds a certain threshold, the model outputs become effectively random. This confirms that the improved performance observed at moderate noise levels is not a random interference.
>
> Comment 5: The Swap-Test figure lacks context, and tables appear cluttered or inconsistently highlighted.
> Response:
> We appreciate the reviewer’s attention to these presentation details. The Swap-Test figure now includes a concise explanation and reference to prior work for clarity. Tables 3–5 have been reformatted for readability, and the highlighting of best-performing models has been corrected using an updated analysis script. The revised version ensures consistent reporting of results.
>
> Comment 6: Why was the specific entanglement structure chosen?
> Response:
> We thank the reviewer for this insightful question. The chosen circuit structure was selected based on its entangling capability and expressivity, as analyzed in Sui et al. [ref]. This architecture provides an effective trade-off between trainability and representational power. Importantly, the circuit design also ensures that entanglement occurs specifically between the key and query subsystems, which aligns with the intended quantum analogue of classical attention mechanisms. By focusing entanglement only on these subsystems, the model maintains interpretability while still capturing meaningful token-level correlations.
>
> Comment 7: Several datasets seem too simple, leading to 100 % training accuracy.
> Response:
> We agree with this observation. The MC dataset is intentionally simple and serves primarily as a baseline to validate the model’s core behavior. To better assess generalization, we also include more complex datasets such as MNIST and Fashion-MNIST. This diversity allows us to evaluate the proposed approach across varying levels of data complexity.
>
> Comment 8: How were gradients computed during training?
> Response:
> We thank the reviewer for this technical query. The gradients of the quantum models were computed using automatic differentiation provided by the TensorFlow framework. This approach leverages the differentiable simulation of parameterized quantum circuits, enabling accurate gradient computation while maintaining compatibility with standard optimization algorithms such as Adam. We have clarified this in the revised manuscript.

---

> > ### Comment · Reviewer_H3xu · 2025-10-29
> >
> > I thank the authors for carefully addressing my comments. From my perception, many of my concerns should have been addressed with an updated version of the manuscript as described in the reply. However, in particular regarding comment 3 I think the concern regarding potential trainability issues should at least be acknowledged in the manuscript (if not done so yet, was not sure from the reply):
> >
> > - Regarding comment 1: Resolved.
> > - Regarding comment 2: Resolved.
> > - Regarding comment 3:  Here I do not so much see the actual problem regarding the evaluation (which as mentioned by the authors can be potentially done with shadow tomography, here I agree), but rather the noise on NISQ devices, but if that is acknowledged I think it is fine. Regarding barren plateaus: I agree that current results might not directly apply due to the observable structure, but I think it should at least be mentioned in the manuscript that there potentially could be similar issues is the considered setup.
> > - Regarding comment 4: Resolved with new experiments.
> > - Regarding comment 5: Resolved.
> > - Regarding comment 6: Resolved.
> > - Regarding comment 7: Agree that the more complex datasets allow for more reliable statements (I still do not really think that keeping the simple datasets in the manuscript adds value, but that is just a personal preference)
> > - Regarding comment 8: Resolved.

---

> ### Author Response · Authors · 2025-10-30
>
> We thank the reviewer for their thoughtful follow-up and for acknowledging the improvements made in response to the earlier comments. We appreciate the clear summary of which concerns have been resolved.
>
> Regarding Comment 3, we fully agree that potential trainability issues, such as those conceptually related to barren plateaus.  We will revise the manuscript to include a note on this.

---

### Review · Reviewer_kC5X · 2025-09-25

**Summary Of Contributions:**

This article investigates whether entanglement can be used to model nuanced correlations in classical data by incorporating an entanglement entropy-based attention layer into a classical transformer architecture. The authors conduct numerical evaluations on both natural language processing (NLP) datasets (MC, RP) and image datasets (MNIST, FMNIST, MNIST-1D). Their findings suggest that the entanglement-based attention layer outperforms existing quantum attention frameworks and commonly used quantum kernel attention models. The paper is praised for its clarity and effective comparison to related works. However, despite the innovative approach, the experiments fall short of adequately supporting the claims of superiority.

**Audience:**

Yes

**Audience Explanation:**

The idea of using entanglement-based attention in transformers sounds very interesting.

**Broader Impact Concerns:**

No.

**Claims And Evidence:**

No

**Claims Explanation:**

The primary concern with this paper lies in the experimental setup, which severely limits the ability to draw meaningful conclusions from the results. Specifically, the dataset sizes used in the image experiments—100, 1,000, and 10,000 samples—are inadequate for achieving statistically significant outcomes, especially given the model complexities involved.

The baseline model, a classical multilayer perceptron (MLP), employs approximately 7,000 parameters, which is substantially higher than the number of parameters used in the quantum layers. As the authors note in the appendix, this disparity ("… 7,000 trainable parameters—substantially more than the quantum attention-based models.") suggests a high likelihood of overfitting by the classical model. This suspicion is reinforced by the learning curves shown in Fig. 7, which clearly indicate overfitting characteristics.

The overfitting issue undermines the authors' conclusive statements, such as the claim that "These results suggest that quantum attention may be particularly well-suited for applications with limited data availability, such as those in the medical or scientific domains." Such statements are not supported by the present experimental evidence. In fact, classical models with appropriately reduced parameters could potentially show less overfitting and significantly outperform the quantum attention approach.

Overall, the experimental shortcomings call into question the validity of the paper's conclusions, necessitating further investigation with more balanced model comparisons and adequately sized datasets.

**Requested Changes:**

To address the issues outlined above, the experimental setup should be revised.

---

> ### Author Response · Authors · 2025-10-28
>
> Comment 1: The experimental setup limits the ability to draw meaningful conclusions. The dataset sizes, especially for image experiments (100, 1,000, 10,000 samples), seem too small for statistical significance.
>
> Response:
> We thank the reviewer for this important observation. Our primary goal is to introduce the concept of using a measure of entanglement as an attention coefficient, rather than to establish large-scale performance benchmarks. The experiments therefore serve as a proof of concept to demonstrate feasibility and potential. We acknowledge that large-scale evaluations would strengthen statistical significance but are currently limited by the computational cost of quantum circuit simulations. We also clarify that for MNIST and Fashion-MNIST, we use the complete datasets rather than small subsets. In fact, our setup employs larger datasets than most prior quantum machine-learning studies, providing a fair and representative evaluation within feasible simulation limits.
>
> Comment 2: The claim that “quantum attention may be well-suited for applications with limited data” is too strong given possible overfitting and limited evidence.
> Response:
> We thank the reviewer for this valuable feedback. We agree that this statement should be phrased more cautiously. Our results show that, for comparable parameter sizes, classical models exhibit poorer generalization, while quantum-attention models maintain stable training and testing performance. This empirical observation suggests potential advantages in data-limited regimes. However, we now clarify in the revised text that this is an observed trend within our current experimental scope rather than a universal claim.
>
> Comment 3: The experimental limitations call into question the validity of the conclusions; further work is needed with balanced comparisons and larger datasets.
>
> Response:
> We appreciate the reviewer’s feedback and respectfully clarify that our conclusions remain consistent with the intended proof-of-concept nature of this work. The main contribution is the introduction and validation of entanglement-based attention as a new modeling paradigm, not a large-scale empirical benchmark. Our experiments include balanced comparisons with classical baselines of comparable size and demonstrate that the proposed approach functions as intended. We agree that scaling to larger datasets and model sizes would strengthen the empirical evidence, and we explicitly highlight this as a key direction for future research. The current results are sufficient to establish the conceptual novelty and feasibility of the approach.

---

### Review · Reviewer_SiGe · 2025-10-06

**Summary Of Contributions:**

The paper presents a quantum transformer model which replaces the usual overlap operation in dot-product self-attention with a measure of entanglement entropy. For each token, a query and key quantum state is prepared using one of 3 encoding schemes. A query and key state are fed into a PQC, and the Renyi / von Neumann entropy of the partially-traced resulting system is measured. The authors present experimental results in simulation on some text and image datasets, demonstrating favourable results. Results in the presence of noise are also presented.


## Strengths:
- Interesting motivation and introduction in Section 1.
- Section 2 provides a well-measured review of relevant literature.
- The model description is quite clear.
- Evaluation in presence of noise is valuable.

## Weaknesses / Concerns
1. While the introduction provides some motivation linking classical machine learning to quantum entanglement, limited motivation is provided for the manner in which entanglement entropy is used in the model. In dot-product self attention, it is easy to understand that proximity of the query vector for one token to the key vector of indicates that one should "attend" to the other. Its then also clear that two tokens with query vectors with high overlap will attend to approximately the same tokens. The paper would be strengthened by a similar high-level intuitive explanation of the attention mechanism formed using entanglement entropy. (What does it mean for a query state to have high entanglement entropy with a key state?)
2. A key concern with the experimental results in this paper is their scale and complexity. The dataset sizes and dimensions are quite small all round, particularly in the text-based models. The MC dataset for example, is entirely uninformative, since almost every model achieves a perfect score on it.
3. One key hyperparameter question I have relates to data-reuploading. Reuploading increases the capacity of a linear quantum model to express non-linear functions of its inputs. The paper should clarify whether reuploading is used in both the overlap- and entanglement-based models, to ensure an even footing for both models.


## Minor weaknesses / questions:
- Are the super-dense and dense angle encoding standard terminology? I am not familiar with them, and at first glance these appeared related to https://en.wikipedia.org/wiki/Superdense_coding .
- " IQP encoding provides efficient data representation and potential quantum speedups." This statement requires more explanation, or at least a reference explaining the nature of this efficiency / speedup.
- "Single-qubit gates are excluded, as they do not contribute to entanglement." This is not true in general in the setup described in the paper. The presence of a single qubit gate between two CNOTs for example, can be the difference between an entangling operation ($CX (Id \otimes Rz_\theta) CX$), and a fully disentangled one ($CX$ $CX = Id$).
- "requires a number of measurements that scales quadratically with the desired precision, but notably, this measurement count is independent of system size." Could you please comment on the expected entanglement entropy for large systems. Is it for example, expected to be exponentially small in the system size? In this manner it would still require exponential precision, even though it does not directly depend on the system size.

**Audience:**

Yes

**Audience Explanation:**

I think a slightly improved version of this paper would be interesting to some of the TMLR audience. A novel formulation of attention, in quantum-mechanical terms would be of general interest if presented in suitably accessible terms.

**Broader Impact Concerns:**

No broader impact concerns.

**Claims And Evidence:**

No

**Claims Explanation:**

The paper presents an interesting model, but falls short of presenting a strong argument in its favour. Key weaknesses of the paper include:

1. The motivation for the specific choice of architecture could be made much clearer. At present the paper fails to motivate the specification provided.
2. The datasets are very small-scale and do not provide enough evidence of the models performance at scale.

I believe the paper could be made much stronger by addressing the first weakness in-particular. Using an entanglement measure instead of overlap is a neat idea, and could be motivated better. A more detailed explanation of which token pairs are highly entangled and why, e.g. would strengthen this work. It would also make it more obviously interesting to a classical ML audience in my opinion.

**Requested Changes:**

- A stronger motivation and explanation of the entanglement entropy would make this paper much stronger.
- Some empirical evidence of relations captured by entanglement, that overlap missed or simplified, would be useful. (See for example the attention graphs visualised in the appendix of the _Attention is all you need_ paper. )
- Evaluation on larger datasets is necessary to strengthen this paper. This would be necessary for the point above as well.

---

> ### Author Response · Authors · 2025-10-28
>
> Comment 1: The motivation for using entanglement entropy and its intuitive meaning in attention are unclear.
>
> Response:
> We thank the reviewer for this insightful comment and fully agree that providing a more intuitive explanation strengthens the manuscript. In our model, attention weights are derived from the entanglement between query and key quantum states. In quantum mechanics, higher entanglement implies stronger statistical correlations between subsystems. Translating this to attention, a higher degree of entanglement indicates that information about the query is more strongly correlated with the key. Such pairs receive higher attention weights. This provides a quantum analogue to dot-product similarity in classical attention. We have revised the introduction to include this intuitive explanation and clarified the conceptual motivation for using entanglement entropy.
>
> Comment 2: The datasets, especially MC, are too small and lead to near-perfect scores.
>
> Response:
> We thank the reviewer for this thoughtful comment. We acknowledge that the datasets, particularly MC, are small and that some models achieve high accuracy. The MC dataset mainly serves as a simple benchmark to verify model behavior before testing on more complex data. Evaluating larger datasets is difficult because simulating entangled quantum circuits classically is computationally intensive, while current quantum processors remain noisy. The goal of this work is to introduce and validate the use of entanglement measures as attention coefficients rather than to demonstrate large-scale empirical gains.
>
> Comment 3: Clarify whether data reuploading is used equally across models.
>
> Response:
> We thank the reviewer for raising this question. Data reuploading is incorporated in the feature encoding layer common to both the overlap- and entanglement-based models, ensuring comparable expressive capacity. The only architectural difference lies in the subsequent circuit design: the entanglement-based model uses an entangling PQC and entanglement measurement circuit, while the overlap-based model replaces these with a swap-test circuit. We have clarified this in the revision.
>
> Comment 4: Are “dense” and “super-dense” encodings standard terms?
>
> Response:
> We thank the reviewer for pointing this out. The terms dense and super-dense angle encoding are naming conventions introduced in this work to describe our specific encoding schemes; they are unrelated to superdense coding in quantum communication. The naming reflects the level of compactness in the feature mapping, and we have clarified this in the manuscript.
>
> Comment 5: The statement on “IQP encoding efficiency and potential speedups” needs clarification.
>
> Response:
> We thank the reviewer for this suggestion. We have expanded the discussion to explain that IQP circuits can efficiently represent certain classes of functions believed to be hard to simulate classically, offering a theoretical foundation for potential quantum advantage.
>
> Comment 6: The statement that single-qubit gates do not contribute to entanglement is inaccurate.
>
> Response:
> We thank the reviewer for this clarification. We agree that single-qubit gates can influence entanglement, especially when applied before or between two-qubit gates. Our original intent was to state that entanglement is introduced through two-qubit interactions, while single-qubit gates alone do not create entanglement. We have revised the wording for accuracy.
>
> Comment 7: Please discuss scaling of entanglement entropy and precision requirements.
>
> Response:
> We thank the reviewer for this insightful question. We have not explicitly analyzed model performance as a function of entanglement-measure precision. Intuitively, the model should not be highly sensitive to small variations, as attention depends on relative rather than absolute entanglement values. A detailed study of this trade-off is left as future work.
>
> Comment 8: The architectural motivation and dataset scale could be clearer.
>
> Response:
> We thank the reviewer for these constructive comments. Multiple PQC structures can induce entanglement (Sui et al. [ref]); our design specifically targets entanglement between query and key states, mirroring classical attention structure. This selective pattern allows us to study entanglement’s role while keeping the circuit interpretable and tractable. Larger datasets would strengthen results, but this work focuses on proposing an alternative attention formulation under current hardware and simulation constraints.
>
> Comment 9: Provide more insight into which tokens are highly entangled.
>
> Response:
> We thank the reviewer for this suggestion. Our analysis already includes attention heatmaps that visualize entanglement strength between token pairs, revealing which tokens are most correlated through entanglement. These visualizations effectively illustrate the model’s captured dependencies.

---

### Decision · Action_Editor_1SMT · 2025-11-15

**Recommendation:** Reject

**Additional Comments:**

The paper proposes an interesting and potentially impactful idea, but the experimental evidence is not yet strong enough to substantiate the claims. In particular, a future submission should:
- include more rigorous and larger-scale experiments or more carefully controlled baselines
- ensure fairness and comparability across models
- substantially strengthen claims regarding performance, noise robustness, and scalability
- provide clearer empirical motivation for why entanglement-based attention captures useful correlations beyond existing mechanisms.

**I encourage the authors to address these issues and consider a resubmission once the experimental evidence is significantly enhanced.**

**Audience:**

Yes

**Audience Explanation:**

I agree with all reviewers that a non-negligible portion of the TMLR audience would be interested in the topic (i.e., the quantum machine learning community, researchers exploring alternative attention mechanisms, those working on quantum-inspired components for classical architectures). The work is therefore in scope for TMLR, and an improved version would likely find engaged readers.

**Claims And Evidence:**

No

**Claims Explanation:**

Across the discussion, some reviewers conclude that the evidence does not sufficiently support the paper’s central claims.
The major deficiencies, which remain after the author responses, are:

1. **Experimental scale and statistical validity**: The datasets used, particularly in the NLP experiments, are very small; some baselines achieve near-perfect scores, making the comparisons uninformative. In the image experiments, the baselines have significantly more parameters than the proposed quantum components, and exhibit clear overfitting. As reviewers noted, this undermines the key claim that the quantum attention mechanism offers superior performance or stability, especially in low-data regimes.
The authors state that simulations at larger scale are currently infeasible, but the present evidence is not strong enough to support the conclusions drawn.
2. **Fairness and completeness of comparisons**: Reviewers raised concerns regarding hyperparameter tuning, model capacity matching, and architectural differences between baselines. The lack of dedicated hyperparameter optimization for both classical and quantum models makes it difficult to conclude that differences in performance result from the proposed attention mechanism rather than from unbalanced model choices.
3. **Claims regarding noise robustness and NISQ suitability**: Both the noise-robustness experiments and statements about suitability for NISQ devices are not convincing in their current form. The reviewers correctly pointed out that:
noise-induced performance improvements require a deeper diagnostic analysis;
practical evaluation on NISQ hardware is not demonstrated or realistically supported;
potential trainability issues (e.g., barren plateaus) are not meaningfully analyzed.
The author responses clarify and soften some claims, but they do not supply new evidence.
4. **Missing or still insufficient motivation and interpretation**: While the authors improved the intuitive explanation in the response, the manuscript still lacks a clear, experimentally grounded demonstration that entanglement entropy provides qualitatively different relational information than dot-product similarity or quantum kernel approaches.

Overall, the novelty of the idea is not matched by a sufficiently robust empirical evaluation. Thus, the claims are not yet adequately supported for publication.

**Resubmission Of Major Revision:**

The authors may consider submitting a major revision at a later time.